# In Silico Analysis of miRNA-Regulated Pathways in Spinocerebellar Ataxia Type 7

**DOI:** 10.3390/cimb47030170

**Published:** 2025-03-02

**Authors:** Verónica Marusa Borgonio-Cuadra, Aranza Meza-Dorantes, Nonanzit Pérez-Hernández, José Manuel Rodríguez-Pérez, Jonathan J. Magaña

**Affiliations:** 1Laboratory of Genomic Medicine, Department of Genetics, Instituto Nacional de Rehabilitation Luis Guillermo Ibarra Ibarra, Mexico City 14389, Mexico; 2Center for Research in Health Sciences, Faculty of Health Sciences, Universidad Anáhuac Mexico Norte, Mexico City 52786, Mexico; 3Department of Bioengineering, School of Engineering and Sciences, Tecnologico de Monterrey, Campus Ciudad de Mexico, Mexico City 14380, Mexico; a01730904@tec.mx; 4Department of Molecular Biology, Instituto Nacional de Cardiología Ignacio Chávez, Mexico City 14080, Mexico; unicanona@yahoo.com.mx (N.P.-H.); josemanuel_rodriguezperez@yahoo.com.mx (J.M.R.-P.)

**Keywords:** spinocerebellar ataxia type 7, polyQ disease, miRNAs, in silico analysis, biomarkers

## Abstract

Spinocerebellar ataxia type 7 (SCA7) is an inherited neurodegenerative disease characterized by cerebellar ataxia and retinal degeneration, caused by an abnormal expansion of the CAG trinucleotide in the coding region of the ATXN7 gene. Currently, in silico analysis is used to explore mechanisms and biological processes through bioinformatics predictions in various neurodegenerative diseases. Therefore, the aim of this study was to identify candidate human gene targets of four miRNAs (hsa-miR-29a-3p, hsa-miR-132-3p, hsa-miR-25-3p, and hsa-miR-92a-3p) involved in pathways that could play an important role in SCA7 pathogenesis through comprehensive in silico analysis including the prediction of miRNA target genes, Gen Ontology enrichment, identification of core genes in KEGG pathways, transcription factors and validated miRNA target genes with the mouse SCA7 transcriptome data. Our results showed the participation of the following pathways: adherens junction, focal adhesion, neurotrophin signaling, endoplasmic reticulum processing, actin cytoskeleton regulation, RNA transport, and apoptosis and dopaminergic synapse. In conclusion, unlike previous studies, we highlight using a bioinformatics approach the core genes and transcription factors involved in the different biological pathways and which ones are targets for the four miRNAs, which, in addition to being associated with neurodegenerative diseases, are also de-regulated in the plasma of patients with SCA7.

## 1. Introduction

SCA7 is an autosomal-dominant neurodegenerative disease characterized by gradual dysfunction of the cerebellum and brain stem, with mainly adult onset of progressive cerebellar ataxia. Therefore, there are different clinical manifestations such as ataxia gait, dysarthria, dysmetria, dysdiadochokinesia, hyperreflexia and postural tremor. In SCA7, irreversible retinopathy is a characteristic feature, often manifesting as the progressive loss of central vision, which ultimately leads to complete blindness [1,2].

SCA7 is caused by an abnormal Cytosine–Adenine–Guanine (CAG) repeat expansion in the *ATXN7* gene coding region located on chromosome 3p12-21.1 [3], which in turn leads to a polyglutamine (polyQ) expansion at the amino terminus of the encoded protein, ataxin-7 (ATXN7) [4]. The polymorphic tract of CAG repeats ranges from 4 to 18 in normal chromosomes and from 36 up to 460 repeats in expanded chromosomes [5]. Interestingly, the length of the CAG tract inversely correlates with age at onset and disease severity [4].

A polyglutamine (polyQ)-tract expansion encoded by the mutant allele imparts aberrant dominant activity to the protein, disrupting multiple cellular processes such as protein folding and clearance, autophagy, oxidative stress and transcriptional regulation [6]. The mutant ATXN7 exhibits toxic gain or loss of function properties, forming aberrant interactions with normal proteins and accumulating in neurons such as intranuclear inclusions, leading to cellular disruption and apoptosis, thereby contributing to SCA7 pathogenesis [7]. This dysfunction is linked to neurodegeneration through impaired protein homeostasis, caused by the accumulation of damaged proteins and the disruption of the ubiquitin–proteasome regulatory system [8]. Additionally, studies in PC12 and HEK 293 cell lines overexpressing mutant ATXN7 have demonstrated reduced autophagic activity due to the sequestration of FIP200, destabilizing the ULK1–ATG13–FIP200 complex essential for autophagy initiation. These findings were corroborated in the Purkinje neurons of SCA7 knock-in mice [9]. In the context of SCA7, a reduction in the coenzyme nicotinamide adenine dinucleotide (NAD+) was observed, leading to pronounced mitochondrial dysfunction [10,11]. Furthermore, alterations in antioxidant systems and an increase in oxidative stress have been reported in both cellular models and patients with SCA7 [12,13,14].

One of the most extensively studied mechanisms in SCA7 is the transcriptional dysregulation of the SPT3-TAF9-Acetyltransferase-GCN5 (STAGA) transcription coactivator complex, which plays a crucial role in chromatin remodeling by mediating histone acetylation and deubiquitination [15]. Consequently, this dysregulation affects the expression of multiple genes [15,16]. For instance, the cone–rod homeobox protein (CRX), a transcription factor essential for retinal photoreceptor development, regulates the expression of key retinal genes. At the transcriptional level, CRX is negatively regulated by the STAGA coactivator complex, leading to persistent alterations in the expression of CRX-regulated genes critical for photoreceptor function. This dysregulation is a contributing factor to retinal dystrophy in SCA7 [3,17]. Moreover, transcriptional dysregulation has been reported in SCA7 knock-in models, suggesting that it is a key mechanism in the molecular pathogenesis of the disease [18].

Studies have demonstrated the presence of miRNAs in different organs, including the brain, where they play a role in neurodevelopment, synaptic plasticity, and microglial activation. However, an imbalance in their expression contributes to neuronal degeneration by inhibiting autophagy, promoting neuroinflammation, neurodegeneration, and apoptosis, among other biological processes in PolyQ diseases [19]. miRNAs act at the post-transcriptional level, regulating gene expression by either silencing or degrading mRNAs. These molecules are endogenous and belong to the class of non-coding RNAs [20,21] and participate in vital cellular processes across different organs and tissues [21,22,23]. The presence of these molecules has been recognized as a potential diagnostic biomarker [24].

Recently, a systematic review conducted by Juźwik et al. reported various dysregulated miRNAs (miR-9-5p, miR-21-5p, the miR-29 family, miR-124-3p, miR-132-3p, miR-146a-5p, miR-155-5p, and miR-223-3p) that are more prevalent in neurodegenerative diseases, including Alzheimer’s disease, Parkinson’s disease, and Huntington’s disease [25]. Specifically, Borgonio et al. reported that hsa-miR-29a and hsa-miR-132-3p were overexpressed in the plasma of patients with SCA7 [26]. The significance of miR-29a lies in its role in proper motor function, as the loss of miR-29a/b function results in severe motor impairment, as reported in an animal model [27,28]. Regarding the importance of miR-132-3p, it promotes neurite (dendrite) growth [29], and the deletion of miR-132-3p induces neuronal apoptosis [30].

Additionally, the miR-25/92 subfamily, a member of the miR-17~92 family, regulates neuronal proliferation and differentiation of neural stem and/or progenitor cells in both developing and adult brains. Its misexpression is associated with acute and chronic neurological disorders by attenuating neurogenesis and repressing neuronal apoptosis [31]. Furthermore, members of the miR-25~92 subfamily (hsa-miR-25-3p and hsa-miR-92a-3p) have been reported as dysregulated in the plasma or post-mortem brain tissue of patients with Huntington’s disease and SCA3, suggesting their potential use as diagnostic biomarkers [32,33,34]. In addition, hsa-miR-25-3p and hsa-miR-92a-3p have been described as overexpressed in SCA7 [26] but their biological significance remains unknown.

In an integrative way, modern medicine has relied on computational sciences (such as bioinformatics) to allow the interpretation of large amounts of biological information, thus providing a better understanding of cellular and molecular mechanisms, interaction pathways, and in silico prediction models in various diseases. It is expected that in the near future, the results obtained from these tools could contribute to the development of new therapeutic strategies, impacting the use of translational medicine, and thus benefiting patients with SCA7.

Therefore, this type of scientific approach helps to shed light on the discovery of possible novel markers in neurodegenerative diseases. In this regard, Hossain et al. provided novel knowledge including the markers, pathways and therapeutic targets involved in the molecular processes of Huntington’s Disease and therapeutic targets using informatic-based analysis and applying network-based systems biology approaches [35,36]. Furthermore, Romano et al. and Jovičić reported an in silico analysis of miRNA regulation in glaucoma and Parkinson’s disease; these results showed valuable information when assessing the de-regulation of specific miRNAs as potential markers and therapeutic targets in other neurodegenerative diseases [37,38].

To our knowledge, there are no studies that address an in silico approach to these four miRNAs that would allow for elucidating possible molecular pathways involved in SCA7. Therefore, the aim of this study was to identify the candidate gene targets of four miRNAs (hsa-miR-29a-3p, hsa-miR-132-3p, hsa-miR-25-3p, and hsa-miR-92a-3p) involved in pathways that could play an important role in SCA7 using integrative in silico analysis.

## 2. Materials and Methods

### 2.1. Exploration of miRNA Families and Selection of miRNAs for the Study

Previously, our group reported miRNA expression data from plasma samples SCA7 patients using Taqman profiling low-density assay (TLDA) microfluidic cards (Human miR ver. 2.0; ThermoFisher Scientific, Melbourne, Australia) that contain a 384-well PCR-based microfluidic card with TaqMan primers and probes in each well for the 380 different mature human miRNA and four nucleolar RNA controls [26]. As a result, 71 miRNAs were found to be significantly upregulated in the plasma of these patients. The confidence of TLDA data was validated by qPCR in 17 SCA7 patients and 12 healthy controls and 4 validated miRNAs were discriminated for diagnosis of SCA7. Given the above, in the present research, we choose the upregulated miRNA of TLDAs and then, we grouped them into different miRNA families aiming to identify miRNA families associated with neurodegenerative diseases.

Therefore, in this study, we focused on the upregulated miRNAs identified through TLDA analysis and grouped them into distinct miRNA families to explore their association with neurodegenerative diseases. From this grouping, four miRNAs (hsa-miR-29a-3p, hsa-miR-132-3p, hsa-miR-25-3p, and hsa-miR-92a-3p) were selected for further analysis due to their potential involvement in neurodegenerative processes.

### 2.2. Gene Ontology and Pathway Enrichment Analysis

To determine the possible involvement of these potential miRNAs in the pathogenesis of SCA7, we used the miRNet 2.0, which is an open-access platform that includes 11 databases of validated RNA target predictions, such as miRTarBase, TarBase, SM2miR, miRanda, phenomiR, miR2Disease, pharmacomiR, miRecords, starBase, EpimiR and HMDD [39]. With support of miRNet 2.0, we performed the prediction of messenger RNAs that target miRNAs. Subsequently, using gene ontology (GO), we performed a functional enrichment analysis through module biological processes (BP). Additionally, we integrated a KEGG (Kyoto Encyclopedia Genes and Genomes) pathway enrichment analysis. The algorithm was assessed using the miRTarBase database, which includes experimentally validated target genes [40]. The analyzed biological pathways were selected considering their relationship with those de-regulated pathways that impact neurodegenerative diseases, with an adjusted *p* < 0.05.

### 2.3. Identification of Hub Genes in KEGG Pathways

Subsequently, through the miRNet platform, we obtained the interaction networks of miRNAs with their target genes, highlighting the nodes as miRNAs and the edges represented by the mRNAs [39]. In addition, in order to identify essential genes in the interaction networks, we used Cytohubba plugin from Cytoscape to show the top 10 hub genes of biological pathways, applying the Maximal Clique Centrality (MCC) as a topological analysis method, which allows us to increase the sensitivity and specificity when selecting the essential genes of each KEGG pathway [41].

### 2.4. Transcription Factors and Hub Genes

The Network Analyst 3.0 tool (available at https://www.networkanalyst.ca/ (accessed on 1 October 2024)) was used to identify the transcription factors (TF) responsible for interacting with hub genes and regulating their expression. This program enables the analysis of relationships between genes and transcription factors from experimental datasets. For this purpose, we use the genes previously identified through the analysis MCC with the plugin cytoHubba, these were introduced for each of the signaling pathways evaluated in the study. We selected Homo sapiens as the reference organism, and the JASPAR database for the transcriptional factor analysis.

The resulting TF-gene regulatory networks were visualized and restructured using Cytoscape 3.10.3. The ten most significant genes were identified by applying the Betweenness Centrality Score methodology, with priority given to those with the highest relevance within the regulatory networks.

### 2.5. Validation of miRNA Target Genes in the SCA7 Mouse Transcriptome

Due to the lack of transcriptomic data in SCA7 patients, and since the disease has been modeled in knock-in mice, we now validated the in silico results shown by the miRNAs evaluated in this animal model.

For this purpose, we considered 2 datasets in GEO database GSE49115 and GSE3634, that were based on Agilent-014868 Whole Mouse Genome Microarray 4x44K G4122F (Agilent Technologies, Santa Clara, CA, USA) and Affymetrix Mouse Expression 430A Array (Thermo Fisher Scientific, Santa Clara, CA, USA), respectively. The sum of both arrays covers the study of mouse cerebellum extracts at 10 days, 22 days and 11 weeks of development with SCA7 and in retinal cells (rods) of R7E-SCA7 transgenic mice, at the onset stage of disease (3 weeks) and at the moderate stage of the disease (9 weeks). In addition, these organs are primarily affected in SCA7. Cerebellar and retinal transcriptome genes (*p* < 0.05 and a Fold Change > 1.5) were overlapped with miRNA target genes.

## 3. Results

### 3.1. Prediction of miRNA Target Genes, Gene Ontology Enrichment and Pathway Analysis

Given the potential neurodegenerative role of the selected miRNA families and their significant dysregulation in the plasma of SCA7 patients, we conducted a detailed analysis of their possible implications using GO enrichment Biological Process (BP) of the hsa-miR-29a-3p, hsa-miR-132-3p, hsa-miR-25-3p, and hsa-miR-92a-3p, and we showed the top 20 pathways of the four miRNAs: negative regulation of protein and RNA metabolic process, negative regulation of transcription from RNA polymerase II promoter, negative regulation of transcription, DNA-dependent, chromatin modification, intracellular protein transport, maintenance of protein location in cell, and y histone modification (Figure 1).

We considered whether the four miRNAs studied could have a common relationship (since different miRNAs can regulate the same messenger RNA and consequently, they could regulate common pathways). Therefore, we analyzed the four miRNAs again using miRNet v 2.0 in order to elucidate their participation in the pathology of SCA7. After using this platform, we identified at least the following eight pathways that can be regulated by the four miRNAs that we identified as being overexpressed in the plasma of patients with SCA7: the focal adhesion pathway with 90 target genes, the neurotrophin signaling pathway with 57 target genes, the protein processing in the endoplasmic reticulum with 58 target genes, adherens junctions with 34 target genes, regulation of the actin cytoskeleton with 72 target genes, RNA transport with 52 target genes, apoptosis with 33 target genes, and dopaminergic synapse with 44 target genes, all important pathways in SCA7 (Appendix A; Figure 2).

It is important to highlight that we also observed pathways like the signaling pathways p53, MAPK, TGF-beta, Wnt, mTOR and Notch, that have been related to neurogenesis and brain disorders at different levels, causing diseases in the adult brain.

### 3.2. Identification of Hub Genes in KEGG Pathways and Transcriptional Factors (TFs)

KEGG pathway enrichment analysis using Cytohubba Plugin Cytoscape showed the top 10 most potent hub genes. In the adherens junction pathway, we identified the genes *ACTB*, *ACTG*, *NLK*, *TGFBR1*, and *TGFBR2*; highlighting *SMAD2*, *CDH1*, *CTNNB1*, *SSX2IP*, *SMAD4* as the most essential genes in this pathway (Figure 3a). On the other hand, transcription factor (TF) prediction analysis showed a complex network of 45 TFs, highlighting FOXC1 as the main regulator of core genes (Appendix A).

Regarding the focal adhesion pathway, our analysis showed the genes *CCND1*, *CRK*, *CRKL*, *CTNNB1*, *FLNA*, *GSK3B*, *ITGB1*, *ITGB8*, *PPP1CC*, and *PTEN*; highlighting *CRKL*, *GSK3B*, and *PTEN* as hub genes (Figure 3b). Furthermore, in silico analysis showed a network of 41 TFs that could regulate the 10 core genes, with the transcription factors FOXC1 and SREBF1 being the relevant ones (Appendix A).

With respect to the neurotrophin signaling pathway, we found the genes *CRKL*, *GSK3B*, *YWHAH*, *JUN*, *KRAS*, *MAPK8*, *NRAS*, *PIK3R1*, *RPS6KA3*, and *YWHAE*; underlining *CRKL* and *YWHAH* as the most essential genes in this pathway (Figure 3c). Furthermore, transcription factor prediction analysis showed a complex network of 40 TFs, highlighting FOXC1 as the main regulator of core genes (Appendix A).

Concerning the protein processing in the endoplasmic reticulum pathway, 8 of the 10 hub genes were top-ranked genes (*BAK1*, *HSPA1B*, *MAPK8*, *NFE2L2*, *SEC23A*, *UBE2D3*, *UBQLN1*, and *XBP1*), unlike the genes *PLAA* and *UBE2G1* (Figure 3d). Regarding the transcription factor analysis, 44 TFs were observed, of which FOXC1 and JUN act as the main regulatory molecules of the core genes (Appendix A).

Moreover, in the regulation of the actin cytoskeleton pathway, the 10 hub genes ranked as the most essential genes were *ARPC2*, *CRKL*, *DIAPH1*, *ITGA6*, *ITGA8*, *ITGAV*, *ITGB8*, *KRAS*, *NRAS*, and *VCL* (Figure 4a). Then, we performed the predictive analysis of transcription factors from the 10 core genes. This analysis showed 42 TFs, highlighting FOXC1 and GATA2 as the main factors (Appendix A).

Regarding the RNA transport pathway, the 10 hub genes were *EE1A1*, *EIF1*, *FXR1*, *GEMIN2*, *GEMIN5*, *KPNB1*, *RAN*, *RANGAP1*, *XPO1* and *XPOT*. However, the four most essential genes in the pathway are *FXR1*, *RAN*, *RANGAP1*, and *XPOT* (Figure 4b). Subsequently, during the in silico analysis, we found a network of 43 TFs that could regulate the 10 core genes, of which the TFs FOXC1 and PPARG are the most relevant (Appendix A).

In the apoptosis pathway of the 10 hub genes (*CASP7*, *FASLG*, *IL1R1*, *IRAK1*, *PIK3CB*, *PIK3CD*, *TNFRSF10B*, *TP53*, *ATM*, *PRKAR1A*) the most essential genes are *ATM* and *PRKAR1A* (Figure 4c). Then, by integrating the transcription factor analysis, we found a complex network of 51 TFs, with FOXC1 standing out as the main regulator of the core genes (Appendix A).

Finally, in the dopaminergic synapse pathway, the analysis showed that the hub genes are *ATF2*, *CREB3L2*, *GNAI3*, *GNAQ*, *GNB1*, *GSK3B*, *KIF5B*, *MAPK8*, *PPP1CC*, and *PPP2R5C* (Figure 4d) and the prediction transcription factor analysis showed 51 TFs, where FOXC1 was the main regulator of the core genes (Appendix A).

### 3.3. Validation of miRNA Target Genes in the Mouse SCA7 Transcriptome

In order to validate the previous results obtained from the in silico analysis where the potential target genes of the four miRNAs were identified, we overlapped the predicted miRNA target genes from eight KEGG pathways with the transcriptome of cerebellum and retina in the mouse model of SCA7. A total of common genes were identified and grouped across the eight KEGG pathways analyzed. In the adherent junction pathways, the genes *CHH1* and *TGFBR2* were highlighted as central genes (Figure 3a). In the focal adhesion pathway, 10 overlapping genes were identified including *TGA3*, *LAMB1*, *ROCK2*, *COL3A1*, *PDGFA*, *VWF*, *XIAP*, *COL4A4*, *CCND2*, and *THSB1* (Figure 3b). The neurotrophin signaling pathway had two overlapping genes, *CAMKA* and *IRS2* (Figure 3c), for the protein processing in the endoplasmic reticulum, six common genes were identified (*UBQLN1*, *SEC61A1*, *TRAM1*, *XBP1*, *DNAJC3*, *EIF2AK1*) with the genes highlighted in red being central to the pathway (Figure 3d). The regulation of actin cytoskeleton showed four overlapping genes (*ITGA3*, *TIAM2*, *ROCK2*, *ENAH*), Figure 4a. While the RNA transport pathway revealed three genes (*EIF3E*, *NUP62*, y *MAGOHB*), Figure 4b. In the apoptosis pathway, three genes were shown (*FLAR*, *XIAP*, and BID), Figure 4c. Ultimately, the dopaminergic synapse pathway showed four genes in common *GNB1*, *PPP2R5C*, *FOS*, and *ITPR*, with the genes highlighted in red being central to the pathway (Figure 4d).

## 4. Discussion

The involvement of different cellular and molecular mechanisms in the onset and progression of SCA7 has been poorly elucidated. Despite the progress in the development of several experimental models, the knowledge of SCA7 is still limited, mainly due to the complex nature and wide intrinsic variability of this disease. Recently, the emergence of new knowledge through the application of omics sciences and bioinformatics integration has been crucial in consolidating strategies of translational medicine to provide a tailored treatment for patients.

Nowadays, there is only one report by Borgonio-Cuadra et al. that describes the possible participation of the FAS signaling pathway, heparan sulfate biosynthesis and SNARE interaction in vesicular transport (which are considered targets of the miRNAs hsa-let-7e-5p, hsa-let-7a-5p, hsa-miR-18a-5p and hsa-miR-30b-5) in patients with SCA7. These pathways are relevant in the gradual loss of specific neurons, as well as in the regulation of cellular homeostasis [26].

In this study, we performed a bioinformatics analysis that included four miRNAs that are de-regulated in the plasma of patients with SCA7 (hsa-miR-29a-3p, hsa-miR-132-3p, hsa-miR-25-3p, and hsa-miR-92a-3p) and have a potential role in neurodegeneration [26]; in addition, a systematic review showed that these four miRNAs were also de-regulated in at least 12 neurodegenerative diseases, highlighting Alzheimer’s, Parkinson’s and Huntington’s diseases among others [25].

The in silico analysis that we performed predicted the possible participation of several common pathways including the focal adhesion pathway, adherens junction pathway, neurotrophin signaling pathway, endoplasmic reticulum processing pathway, actin cytoskeleton regulation, RNA transport, apoptosis, and dopaminergic synapse that could play an important role in SCA7 and are consistent with previous knowledge in other neurodegenerative diseases. Therefore, addressing these pathways and their interrelation is a viable field for exploration, since evidence of the involvement of these pathways is scarce. Furthermore, the identification of essential genes housed in the core genes in each pathway provides new knowledge to address the disease.

It is important to consider that the pathways identified in our study and that will be described are related to the pathways in other neurodegenerative diseases, so their participation and relevance are known. However, in SCA7, there are few findings of these pathways impacted by the miRNAs hsa-miR-29a-3p, hsa-miR-132-3p, hsa-miR-25-3p and, hsa-miR-92a-3p. Therefore, they represent possible additional mechanisms for a better understanding of the disease.

Recent studies have shown that focal adhesion pathways and the adherens junction form macromolecular structures in order to anchor and maintain cohesion in many tissues such as epithelia, muscle and nervous tissues. In the central nervous system, for instance, various molecules involved in the adhesion of nerve cells regulate the stabilization of synaptic junctions, neurite growth, neuronal plasticity, and the controlled release of enzymes related to the biosynthesis of neurotransmitters [42]. However, dysfunction of cell adhesion molecules could influence the de-regulation of the health–disease balance. Also, the adherens junction pathway has been described to play a crucial role in the regulation, growth and development of the retina in Drosophila, determining the basal–apical polarity of retinal cells as well as controlling the shape of photoreceptors [43]. The alteration of the molecules involved in adherens junctions as well as other types of cellular junctions have been related to the alteration of the blood–retinal barrier and this alteration facilitates the onset and progression of retinopathy [44,45].

Furthermore, genes of relevance in the adherens junction pathway, catenin beta 1 (*CTNNB1*), E-cadherin (*CDH1*), beta actin (*ACTB*), and gamma ACTIN (*ACTG1*) could be of relevance in future studies due to their importance in the pathway. In SCA7, one of the clinical features is retinopathy pigmentosa, which is characteristic of patients with more severe disease and the cone–rod dystrophy phenotype in patients is observed.

Interestingly, the connection between *PTEN* (phosphatidylinositol-3,4,5-trisphosphate 3-phosphatase), one of the central genes in the focal adhesion pathway, and *FAK* has been demonstrated. PTEN protein dephosphorylates FAK and inhibits cell migration, spreading, and focal adhesion formation [46]. Moreover, TGF-β receptors (TGFBR1, TGFBR2) and second messengers (SMAD2, SMAD4) act as targets of miRNAs, which we have identified among the most essential genes in the adherens junction pathway. Also, aging and chronic inflammation have an impact on the reduction of canonical TGF-β1/Smad signaling, causing cytotoxic activation of microglia and microglia-mediated neurodegeneration. This evidence has conferred a neuroprotective role of TGF-β in Alzheimer’s disease; therefore, re-establishing the alteration of TGF-β1 signaling could be an alternative in the treatment for this and other neurodegenerative diseases [47]. Therefore, similar mechanisms could be associated with SCA7.

Another signaling pathway that is possibly regulated by the miRNAs hsa-miR-29a-3p, hsa-miR-132-3p, hsa-miR-25a-3p and hsa-miR-92a-3p is the neurotrophin pathway; neurotrophins were initially identified as molecules related to neuronal survival, that is, they are recognized for their wide range of physiological responses apart from survival and apoptosis, including the regulation of tissue proliferation, regulation of neurite growth, modulation of synaptic responses [48].

On the other hand, it is known that neurotrophins are growth factors that have a pleiotropic effect through signal transduction in different target cells. Members of the neurotrophin family include the neural growth factor (NGF), neurotrophin 3 (NT3), and neurotrophin 4 (NT4/5), and brain-derived neurotrophic factor (BDNF). The participation of the molecules in the development, survival and maintenance of neural cells has been described; nonetheless, de-regulation or modification in the levels of these growth factors have been identified in neurodegenerative processes [49]. Importantly, according to our analysis, we can highlight that hsa-miR-132-3p could regulate the expression of the neurotrophic molecule BDNF effect [50]. In addition, we highlight that CRK-like protein (CRKL) mediates intracellular signal transduction, and Tyrosine 3-Monooxygenase/Tryptophan 5-Monooxygenase Activation Protein Eta (YWHAH) involved in signal transduction [51,52].

Furthermore, previous studies have demonstrated the key role of BDNF in cerebellar structure and function through mice lacking BDNF showing reduced dendritic arborization of cerebellar Purkinje cells, loss of synapses, and impaired short- and long-term synaptic plasticity [53,54]. In addition, BDNF regulates hippocampal neurogenesis and cognition [55,56]. Also, brain disorders, including psychiatric and neurodegenerative diseases, are often associated with decreased BDNF levels in the brain [57,58,59].

It has also been observed that various physiological and pathological conditions can alter the correct protein processing in the endoplasmic reticulum (ER), this disruption favors the accumulation of misfolded proteins in the endoplasmic lumen, and when misfolded proteins accumulate within the ER leading to prolonged cellular stress, the apoptosis processes activate [60].

Reports have demonstrated that neuronal cells are particularly sensitive to protein misfolding and, consequently, endoplasmic reticulum dysfunction has been described as being involved in different neurodegenerative diseases, including some spinocerebellar ataxias [61,62]. However, this pathway is poorly studied in neurodegenerative diseases, based on our study, using in silico analysis, we identified several molecules involved in this process BAK1, HSPA1B, MAPK8, NFE2L2, SEC23A, UBE2D3, UBQLN1 and XBP1 which could play an important role in SCA7.

A well-described mechanism is mutant ATXN7 (mATXN7), which is a progressive accumulation of misfolded forms of the protein in the brain. Progressively, misfolded mATXN7 forms large aggregates in nuclear inclusions (NI) [63,64], which are distributed in different degenerative brain regions [65], leading to an indicator of mATXN7 toxicity. Evidence from transgenic (tg) and knock-in (KI) mice models of SCA7 suggests that misfolded mATXN7 accumulates in the nuclei of vulnerable neurons such as photoreceptors and Purkinje cells [63,64,66].

Regarding the actin cytoskeleton regulatory pathway, which plays a critical role in regulating cell morphology, signaling and other factors involved in pathological conditions, it could be mediated by the miRNAs hsa-miR-29a-3p, hsa-miR-132-3p, hsa-miR-25a-3p, and hsa-miR-92a-3p. Recently, a relationship between the cytoskeletal architecture and alterations due to oxidative stress, or various impaired transport mechanisms such as the ubiquitin–proteasome system (UPS) has been described, which involves the degradation of misfolded cellular proteins that impact several neurodegenerative diseases including Alzheimer’s, Parkinson’s, Huntington’s disease, among others [48,49]. In the case of a Drosophila model, it was shown that loss of cytoskeletal structure leads to defects in axonal transport, deficits in synaptic maturation, and retraction of synaptic buttons [67], suggesting that similar processes could be involved in human neuronal diseases. Therefore, this pathway has a key role in actin on synaptic function and the fact that dysregulation of this cytoskeleton component is emerging in neurodegenerative diseases and the impact of Atnn7 on actin is relevant.

In addition, recent evidence indicates that RNA transport is altered in several neurodegenerative diseases, including Huntington’s disease, frontotemporal dementia, and amyotrophic lateral sclerosis. Studies have described three different alterations in RNA transport: (a) alterations in the membrane, (b) alterations in the Ran GTPase gradient, and (c) alterations in the proteins necessary for mRNA export, leading to nuclear accumulation of mRNA [68].

Furthermore, Ziff et al. reported a widespread redistribution of mRNA, as well as RNA-binding proteins localized in ALS motor neurons, providing evidence of the pathogenesis of the disease with a promising therapeutic target [69]. In our study, we highlight the participation of the *FXR1* gene that encodes an RNA-binding protein involved in posttranscriptional mRNA regulation. FXR1 emerges as an important regulator of RNA metabolism in the brain, with strong implications in neurodevelopmental and psychiatric disorders [70]. Also, we found the *RAN* gene, which is a G protein involved in the nucleocytoplasmic transport of proteins within the nucleus of RNA within the cytoplasm. Additionally, the *RANGAP1* gene encodes a RAN-binding protein that regulates nuclear transport, and the *XPOT* gene encodes for a protein belonging to the exportin RAN-GTPase family that mediates the export of tRNA from the nucleus to the cytoplasm mechanisms that together could be related to retinal neurodegeneration [71].

Another process that has been implicated in neurodegenerative diseases is the apoptosis pathway. In particular, studies carried out by Yefimova et al. in a murine model of SCA7, as well as in an in vitro inducible neuron cell model in PC12 cells, showed that the mutant protein ATXN7 activates the apoptotic pathway in response to cellular stress [72]. However, these results are controversial; in the murine model, an overexpression of p53 has been reported, which promotes the transcription of pro-apoptotic genes (*BAX*, *PUMA*). While in the inducible PC12 cell model, it was observed that the transcriptional activity of p53 is reduced [12,73]; therefore, additional evidence is necessary to achieve a better understanding of this pathway in SCA7.

Through our in silico analysis, we identified genes such as *ATM*, which plays a critical role in regulating cell division and DNA damage repair, as potentially significant contributors to apoptosis. Similarly, *PRKAR1A* (protein kinase A regulatory subunit) was found to promote cell growth and division. Notably, this gene regulates caspase-7 (*CASP7*), an enzyme essential in this pathway, as it may contribute to the pathogenesis of SCA7 through the proteolytic processing of the ATXN7 [74].

Another pathway involved is the dopaminergic synapse pathway. Dopaminergic neurons have been described as participating in important biological processes such as movement, motivation, and intellectual function. Research in humans, both in clinical and experimental settings, has shown that an altered dopaminergic function may be associated with numerous neurological and psychiatric diseases, including movement disorders, schizophrenia, attention deficit hyperactivity disorder, depression, and drug abuse. Several neurological and psychiatric disorders can be considered synaptopathies, that is, diseases caused by dysfunctions in synapses [75,76]. This pathway is possibly related to movement disorders in patients with SCA7.

Furthermore, the presence of postsynaptic dysfunctions with a reduced binding of the dopamine striatal receptor (D1 and D2) has been demonstrated in patients with Huntington’s disease [77]. Also, in HD, it was described that the expression of the mutant Huntingtin protein changes the excitatory synaptic activity of the striatum by decreasing the glutamate uptake and increasing the signaling of N-methyl-d-aspartate receptors (NMDAR) [77]. Various studies indicate that the reduced transcription, transport and signaling of the brain-derived neurotrophic factor (BDNF), contribute significantly to striatal neuronal dysfunction and degeneration in HD [78]. In addition, it highlighted essential genes in the pathway such as *CREB3L2*, *GSK3B*, *MAPK8*, and *PPP1CC*.

It is also important to highlight the possible participation of the transcription factor FOXC1; its participation in the early development of structures such as the lens, iris and cornea has been described [79,80]. Also, in the adult eye, FOXC1 is predicted to maintain homeostasis by regulating genes that are important for stress response. However, FOXC1 dysfunction contributes to the death of cells in the trabecular meshwork of the eye, an important step in the development of glaucoma [81]. Therefore, given its relevance, its study in SCA7 is of great interest.

On the other hand, as perspectives in the present study and considering the importance of achieving a better understanding of the pathways found with SCA7, we propose the functional validation of the identified genes and their effects on the highlighted signaling pathways, ultimately contributing to the identification of molecules with potential therapeutic impact.

In this regard, studies using an in vitro cellular model of SCA7 could assess the overexpression or silencing of candidate miRNAs and monitor changes in the expression of their potential target genes using reverse transcription-polymerase chain reaction (RT-PCR). For example, the expression of Beta-catenin and E-cadherin in the focal adhesion pathway could be validated, as these molecules play a crucial role in cellular cohesion and intracellular communication. The loss of these functions could lead to apoptosis, ultimately resulting in the degeneration of retinal cells and neurons, as documented in both SCA7 patients and animal models.

Furthermore, if Beta-catenin and E-cadherin expression were regulated by any of the identified miRNAs, a decrease in these molecules would likely affect their function, thereby impacting the pathways in which they are involved. With this prior knowledge, it would be possible to explore whether these target molecules could have a therapeutic effect, considering their abundance, deficiency, and relevance in the affected tissue.

Additionally, an in vitro approach could validate the use of synthetic anti-miRNAs to block specific miRNAs, aiming to restore the function of their target genes and potentially influence disease progression. Conversely, if gene silencing were required, strategies such as adeno-associated virus (AAV)-mediated gene silencing via RNA interference or CRISPR-Cas could be employed.

We recognize that our study has some limitations. First, while enrichment analysis based on statistical associations highlighted potential molecular mechanisms involved in SCA7, further functional investigations are essential to confirm their relevance to the disease’s pathobiology.

Second, it is essential to recognize that the interspecific differences between mouse models and humans may affect the study’s findings; this could be due to factors such as regulatory differences, like alternative splicing, chromatin structure, and enhancer activity, among others, that can alter expression patterns in mice and humans. Furthermore, while many of the protein-coding genes possess a high level of conservation, the case may vary in non-coding elements, which may become a discrepancy in disease modeling [82].

Third, an initial bias can be observed from the selection of the database as KEGG, as each contains different representations of the same biological pathway which can be seen represented as a variable result of the enrichment analysis and the contextualization thereof [83]. One should also consider the frequency of platform updates that may not yet cover newly discovered pathways or important updates related to emerging molecular mechanisms, thus truncating the ability for a comprehensive view of biological systems [84].

However, despite the above, our study has several strengths: (a) we assessed an integrative bioinformatic analysis with a variety of tools including prediction of miRNA target genes, Gene Ontology enrichment and pathways analysis, identification of core genes in KEGG pathways, transcription factors and hub genes, we validated miRNA target genes with the mouse SCA7 transcriptome data; this had to be carried out this way since there is no transcriptome data in humans. Therefore, a near perspective will be to evaluate the expression of these miRNA families in patients with SCA7 and assess their expression, aiming to relate the changes in expression to the natural history of the disease, and even to the severity of the pathology.

## 5. Conclusions

Based on a bioinformatics analysis, the present study shows possible new evidence of different biological pathways involved in SCA7. We highlighted a total of eight biological pathways (adherents junction pathway, focal adhesion pathway, neurotrophin signaling pathway, endoplasmic reticulum processing pathway, actin cytoskeleton regulation, RNA transport, apoptosis and dopaminergic synapse). These findings contribute to new knowledge in the bioinformatic field involved in the pathogenesis of SCA7. However, the altered biological processes remain an early recognition stage; experimental validation is essential to confirm and further explore findings.

## Figures and Tables

**Figure 1 cimb-47-00170-f001:**
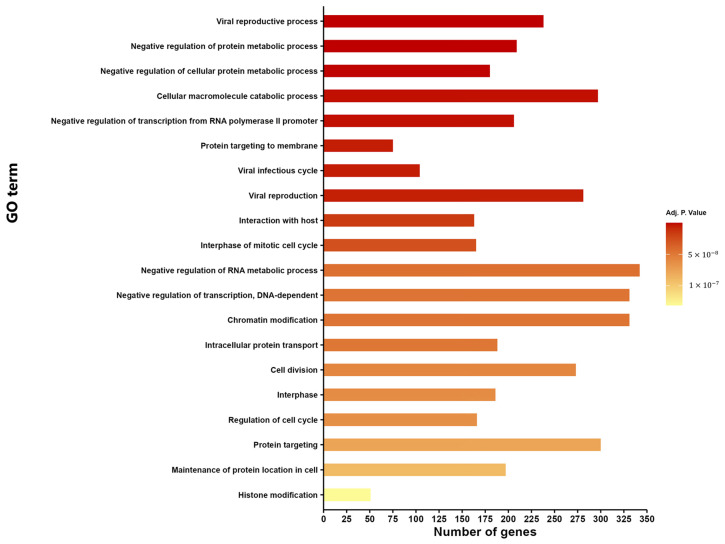
Top 20 Gene Ontology (GO) terms under biological process. The enrichment analysis of potential pathways of hsa-miR-29a-3p, hsa-miR-132-3p, hsa-miR-25a-3p, hsa-miR-92a-3p. Statistically significant differences are shown with an adjusted *p* < 0.05. The *x*-axis represents the number of target genes for miRNA in each pathway. The *y*-axis represents the enrichment GO term.

**Figure 2 cimb-47-00170-f002:**
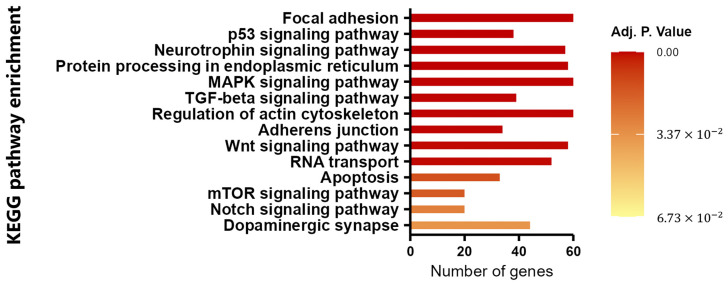
KEGG pathway enrichment analysis of miRNA target genes hsa-miR-29a-3p, hsa-miR-132-3p, hsa-miR-25a-3p, hsa-miR-92a-3p. Fourteen enriched pathways are presented (*p* < 0.05). The *x*-axis represents the number of target genes in each pathway. The *y*-axis represents the KEGG pathways.

**Figure 3 cimb-47-00170-f003:**
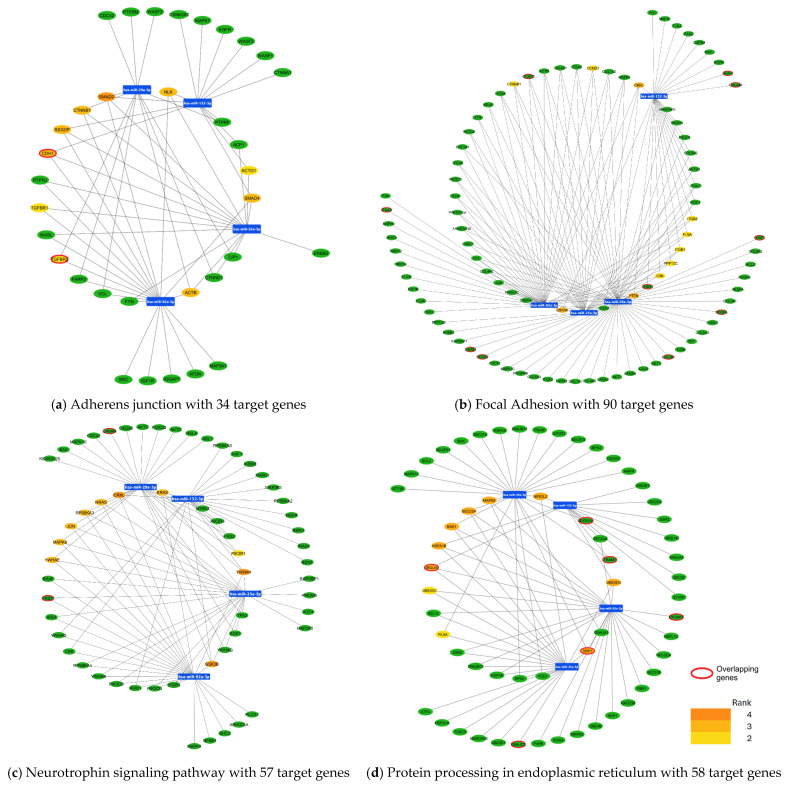
miRNA target genes of hsa-miR 29a-3p, hsa-miR-132-3p, hsa-miR-25a-3p and hsa-miR-92a-3p involved in KEGG pathways. The KEGG pathway shows target genes of miRNAs obtained using the miRNet 2.0 platform. In each pathway, the top 10 hub genes are shown, and we highlight genes de-regulated in the transcriptome of the mouse model for SCA7 that are target molecules of the 4 miRNAs. Panel (**a**) shows target genes in adherens junction. Panel (**b**) shows target genes in the focal adhesion pathway. Panel (**c**) shows target genes in the neurotrophin signaling pathway. Panel (**d**) shows target genes in the protein processing in endoplasmic reticulum. In the bottom-right corner, the intensity of the colors of hub genes shows the ranking position: the dark orange genes have the most significant Maximal Clique Centrality (MCC) values and thus are hub genes of greater importance in the network; the light-yellow ones have lower MCC values and thus are hub genes of lower importance in the network. Genes outlined in red represent the overlapping genes between those identified as de-regulated in the transcriptome of the mouse-SCA7 model and those predicted in silico for each pathway.

**Figure 4 cimb-47-00170-f004:**
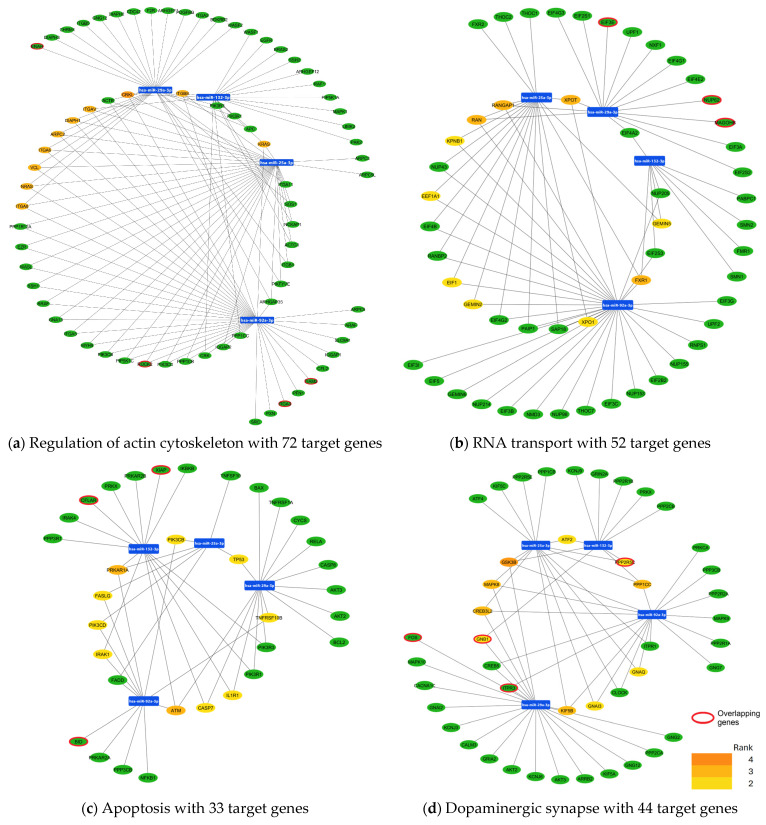
miRNA target genes of hsa-miR 29a-3p, hsa-miR-132-3p, hsa-miR-25a-3p and hsa-miR-92a-3p involved in KEGG pathways. The KEGG pathway shows target genes of miRNAs obtained using the miRNet 2.0 platform. In each pathway, the top 10 hub genes are shown, and we highlight genes de-regulated in the transcriptome of the mouse model for SCA7 that are target molecules of the 4 miRNAs. Panel (**a**) shows target genes in the actin cytoskeleton. Panel (**b**) shows target genes in the RNA transport pathway. Panel (**c**) shows target genes in the apoptosis pathway. Panel (**d**) shows target genes in the dopaminergic synapse pathway. In the bottom-right corner, the intensity of the colors of hub genes indicates their ranking position: the dark orange genes have the most significant Maximal Clique Centrality (MCC) values and thus are hub genes of greater importance in the network; the light-yellow ones have lower MCC values and thus are hub genes of lower importance in the network. Genes outlined in red represent the overlapping genes between those identified as de-regulated in the transcriptome of the mouse-SCA7 model and those predicted in silico for each pathway.

## Data Availability

Data are contained within the article and Appendix A.

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
