# Peer review of "In Silico Analysis of miRNA-Regulated Pathways in Spinocerebellar Ataxia Type 7"

_cimb, 2025, doi:10.3390/cimb47030170_

Round 1

Reviewer 1 Report (New Reviewer)

Comments and Suggestions for Authors

cimb-3431946 entitled In silico identification of pathways regulated by relevant miRNAs in spinocerebellar ataxia type 7, by Verónica Marusa Borgonio-Cuadra and collabbourators, studies the candidate human genes target of 4 miRNAs (hsa-miR-29a-3p, hsa-miR-132-3p, hsa-miR-25-3p, and hsa-miR-92a-3p) involved in pathways that could play an important role in SCA7 pathogenesis through comprehensive in silico analysis. An integrative bioinformatic approach was employed, using a variety of tools including prediction of miRNA target genes, Gen Ontology enrichment and pathways analysis, identification of core genes in KEGG pathways, transcription factors and hub genes. In addition, we validated miRNA target genes with the mouse SCA7 transcriptome data.

The participation of focal adhesion pathway, adherens junction pathway, neurotrophin signaling pathway, endoplasmic reticulum processing pathway, actin cytoskeleton regulation, RNA transport, apoptosis and dopaminergic synapse. These findings contribute to new knowledge in the bioinformatic field involved in the pathogenesis of SCA7. However, the altered biological processes are still in a recognition stage and experimental validation is needed to confirm our findings.

The work is scientifically focused on the pathogenesys of SCA7, an inherited autosomic dominant disease.

The introduction is focused and summarise the state of the art.

Methodology is appropriated.

Results are clear and significant.

Tables and figures are informative.

The discussion is consistent with results.

References are appropriated.

English is good.

Line 252: the notes to figure 2 should stay in the same page of the figure.

Line 304: the notes to figure 3 should stay in the same page of the figure.

Line 315: the notes to figure 4 should stay in the same page of the figure.

Line 370: the notes to figure 5 should stay in the same page of the figure

Line 536: remove interestingly

Author Response

REVIEWER 1

We thank to the reviewer for the insightful comments and helpful criticisms. We have responded to all comments as detailed below and we now hope that you will find our revised Manuscript acceptable for publication in “Current Issues in Molecular Biology”.

REVIEWER'S COMMENTS:

The work is scientifically focused on the pathogenesis of SCA7, an inherited autosomic dominant disease.

The introduction is focused and summarise the state of the art.

Methodology is appropriated.

Results are clear and significant.

Tables and figures are informative.

The discussion is consistent with results.

References are appropriated.

English is good.

Line 252: the notes to figure 2 should stay in the same page of the figure.

Line 304: the notes to figure 3 should stay in the same page of the figure.

Line 315: the notes to figure 4 should stay in the same page of the figure.

Line 370: the notes to figure 5 should stay in the same page of the figure

Line 536: remove interestingly

Answer:

We thank the reviewer for the time invested in quality scientific review and for his valuable contributions to our work.

Following up on his comments, we inform him that we have carefully addressed his 5 clarifications in the revised version of the manuscript.

Thank you again for your invaluable support.

Reviewer 2 Report (New Reviewer)

Comments and Suggestions for Authors

In silico identification of pathways regulated by relevant miRNAs in spinocerebellar ataxia type 7

The manuscript´s aim is to study and identify candidate human genes target of 4 miRNAs (hsa-miR-29a-3p, hsa-miR-132-3p, hsa-miR-25-3p, and hsa-miR-92a-3p) involved in pathways that could play an important role in SCA7 pathogenesis through comprehensive in silico analysis. The authors have utilised integrative bioinformatic approach using a variety of tools including prediction of miRNA target genes, Gen Ontology enrichment and pathways analysis, identification of core genes in KEGG pathways, transcription factors and hub genes. In addition, they validated miRNA target genes with the mouse SCA7 transcriptome data. Their results showed the participation of focal adhesion pathway, adherens junction pathway, neurotrophin signaling pathway, endoplasmic reticulum processing pathway, actin cytoskeleton regulation, RNA transport, apoptosis and dopaminergic synapse. These findings have contributed to new knowledge in the bioinformatic field involved in the pathogenesis of SCA7. However, the altered biological processes are still in a recognition stage and experimental validation is needed to confirm our findings.

The manuscript overall is very well written but I have concerns that would help to improive the qulaity of the manuscript scignificantly.

Major concerns :

a)     The title seems to be clear but overly specific. Simplifying the title to capture broader interest while retaining the focus on spinocerebellar ataxia type 7 (SCA7) may increase appeal. The authors might consider phrasing like: “In Silico Analysis of miRNA-Regulated Pathways in Spinocerebellar Ataxia Type 7.”

b)    The abstract concisely summarizes the study’s aim, methodology, and key findings. But the abstract lacks a clear statement on the novelty of the study. The authors should emphasize how this study differs from prior work in identifying miRNA pathways in SCA7. Also the last statement can be disregarded or removed as it doesn’t really tell the story of the current work. This might be considered in the conclusion or futre aspect of the study. It would be beneficial for the authors to highlight the potential translational impact of the findings.

c)     While going through the intro part, it provides a comprehensive background on SCA7, miRNA functions, and the use of bioinformatics in neurodegenerative diseases. On the opther hand, the introduction does not provide sufficient context on prior in silico analyses of miRNA in related neurodegenerative diseases, which could help situate this study within the broader research landscape.

d)    It is unclear why only four miRNAs were selected, especially since 71 miRNAs were initially identified. A more rigorous selection criterion should be detailed. It would be nice to justify the selection of the four miRNAs with a clear rationale, linking their known roles in neurodegeneration or prior experimental evidence.

e)     The reliance on mouse transcriptomic datasets (GEO) for validation is a significant limitation. These data might not fully represent human SCA7 pathophysiology.

f)     Address how potential interspecies differences between mouse models and humans might affect the study’s findings. This was lacking significantly while reading through ther manuscript. The authors also failed to describe any limitations in using KEGG pathway analysis for predictive modeling.

g)    The authors have described the results section very well and have comprehensively presented bioinormatic analyses. Identification of specific pathways (e.g., focal adhesion, neurotrophin signaling) adds mechanistic insight into SCA7 pathology. But looking at the data the presentation of results seems overwhelming, with excessive details on GO terms and pathway components. This literally obscures the primary takeaways.

h)    The functioanl interepretation is lacking significantly, while enriched pathways are identified, their functional implications for SCA7 are not deeply discussed. The Figures and tables lack clarity and do not effectively summarize the findings.

i)      I felt that the figures are very much overlly representated. It can be shorted from 6 figures to 4 figures. The figures 3-6 looks same represenation of figures, if they have used bioinformatic approach then representing these data in diiferent way would attract readers otherwise they will surely loose interest. It definitely needs to be changed.

Minor concerns

a)    The discussion ties the results to broader neurodegenerative disease mechanisms. It also potentially links between miRNAs and pathological pathways like protein processing and actin cytoskeleton regulation are explored. But there are many assertions which seems to be speculative, with limited experimental evidence to support the proposed links between miRNAs and pathways. It often reiterates known findings without highlighting the unique contributions of this study. The authors might acknowledge the speculative nature of some claims and propose follow-up experiments to validate them. Also they could emphasize novel insights from this study, such as the identification of specific hub genes or transcription factors.

b)    Tables lack summary statistics and clear rankings to prioritize findings.

c)     Figures are way to dense and difficult to interpret. For example, the transcription factor-gene interaction networks (figure 3-6) are cluttered and lack explanatory captions.

Author Response

REVIEWER 2

We thank to the reviewer for the insightful comments and helpful criticisms. We have responded to all comments as detailed below and we now hope that you will find our revised Manuscript acceptable for publication in “Current Issues in Molecular Biology”.

REVIEWER'S COMMENTS:

The manuscript overall is very well written but I have concerns that would help to improve the quality of the manuscript significantly.

Major concerns:

a) The title seems to be clear but overly specific. Simplifying the title to capture broader interest while retaining the focus on spinocerebellar ataxia type 7 (SCA7) may increase appeal. The authors might consider phrasing like: “In Silico Analysis of miRNA-Regulated Pathways in Spinocerebellar Ataxia Type 7”.

Answer: We thank to the reviewer for this suggestion. We totally agree in this important point and we changed the manuscript title as: “In Silico Analysis of miRNA-Regulated Pathways in Spinocerebellar Ataxia Type 7”. (LINES 2 - 3).

b) The abstract concisely summarizes the study’s aim, methodology, and key findings. But the abstract lacks a clear statement on the novelty of the study. The authors should emphasize how this study differs from prior work in identifying miRNA pathways in SCA7. Also, the last statement can be disregarded or removed as it doesn’t really tell the story of the current work. This might be considered in the conclusion or future aspect of the study. It would be beneficial for the authors to highlight the potential translational impact of the findings.

 Answer: Following this pertinent recommendation, the abstract was substantially improved by highlight the main novelty of our study. We now provide in “Abstract” section in the next sentence:

In conclusion, unlike previous studies, we highlight through a bioinformatics approach core genes and transcription factors involved in different biological pathways and which are targets for the 4 miRNAs; which, in addition to being associated with neurodegenerative diseases, are also dereg-ulated in the plasma of patients with SCA7.” (LINES 27 - 31).

In addition, the statement of abstract “These findings contribute to new knowledge in the bioinformatic field involved in the pathogenesis of SCA7. However, the altered biological processes are still in a recognition stage and experimental validation is needed to confirm our findings” was removed of the abstract section and now it was placed in conclusion (LINES 549 - 552).

We hope its current form satisfices the reviewer.

c) While going through the intro part, it provides a comprehensive background on SCA7, miRNA functions, and the use of bioinformatics in neurodegenerative diseases. On the other hand, the introduction does not provide sufficient context on prior in silico analyses of miRNA in related neurodegenerative diseases, which could help situate this study within the broader research landscape.

Answer: We approached this meaningful recommendation in the revised Manuscript. We now include brief information of the context on prior in silico analysis of miRNAs in associated neurodegenerative diseases, based on a searching of published data sets. In addition, we added the 4 mentioned references, which are now references 35, 36, 37 and 38.

Therefore, this type of approach scientific help to shed light on the discovery of possibles novel markers in neurodegenerative diseases. In this regard, Hossain et al. provide novel knowledge including markers, pathways and therapeutic targets involved in the molecular processes of Huntington Disease and therapeutic targets using informatic based analysis and applying network-based systems biology approaches [35, 36]. Furthermore, Romano et al. and Jovičić reported an in silico analysis of miRNAs regulation in glaucoma and Parkinson’s disease; these results showed valuable information regarding to assess deregulation of specific miRNA, as potential markers and therapeutic targets in other neurodegenerative diseases [37, 38]”. (LINES 112 -120).

Moreover, 4 references were added:

  1. Hossain, M.R.; Tareq, M.M.I.; Biswas, P.; Tauhida, S.J.; Bibi, S.; Zilani, M.N.H.; Albadrani, G,M.; Al-Ghadi, M.Q.; Abdel-Daim, M.M.; Hasan, M.N. Identification of molecular targets and small drug candidates for Huntington's disease via bioinformatics and a network-based screening approach. J. Cell Mol. Med. 2024, 28, e18588.
  2. Christodoulou, C.C.; Papanicolaou, E.Z. Integrated Bioinformatics Analysis of Shared Genes, miRNA, Biological Pathways and Their Potential Role as Therapeutic Targets in Huntington's Disease Stages. Int. J. Mol. Sci. 2023, 24, 4873.
  3. Romano, G.L.; Platania, C.B.; Forte, S.; Salomone, S.; Drago, F.; Bucolo, C. MicroRNA target prediction in glaucoma. Prog. Brain Res. 2015, 220, 217-240.
  4. Jovičić, S.M. Analysis of total RNA as a potential biomarker of Parkinson's disease in silico. Int. J. Immunopathol. Pharmacol. 2025, 39, 1- 47.

d) It is unclear why only four miRNAs were selected, especially since 71 miRNAs were initially identified. A more rigorous selection criterion should be detailed. It would be nice to justify the selection of the four miRNAs with a clear rationale, linking their known roles in neurodegeneration or prior experimental evidence.

Answer: We included this valuable information according to the reviewer’s suggestion in a in the introduction section (LINES 76 - 104). In addition, we now clearly describe in “Material and Methods” section (2.1 Exploration of miRNA families and selection of miRNAs for the study) a detailed description of miRNAs selection criteria. (LINES 127 – 143).

We hope its current form satisfices the reviewer.

e)     The reliance on mouse transcriptomic datasets (GEO) for validation is a significant limitation. These data might not fully represent human SCA7 pathophysiology.

Answer: We agree with the reviewer that mouse data may not fully represent the pathophysiology of human SCA7. However, there are no human transcriptomic data on SCA7. Despite the above, the murine model in the actuality remains a crucial tool for understanding and investigating biological mechanisms, particularly at the level of pathways and networks, where studies in mice provide key insights into potential gene regulatory interactions and their functional consequences upon gene dysregulation, which can subsequently be validated in humans. In this regard, we have recognize this issue in the manuscript as a limitation in the discussion section (LINES 522 – 525).

f)     Address how potential interspecies differences between mouse models and humans might affect the study’s findings. This was lacking significantly while reading through their manuscript. The authors also failed to describe any limitations in using KEGG pathway analysis for predictive modeling.

Answer: We thank the reviewer for this comment. Although the mouse (Mus musculus) as a model can be considered a tool of high scientific value thanks to its genetic and physiological homology with humans, it is essential to recognize the interspecific differences between them that can lead to important limitations. In order to clarify this issue, we added the following sentence in the discussion sections as additional limitations:

It is essential to recognize that the interspecific differences between mouse models and humans may affect the study’s findings, this could be due to factors such as regulatory differences, like alternative splicing, chromatin structure, and enhancer activity, among others; that can alter expression patterns in mice and humans. Furthermore, while many of the protein-coding genes possess a high level of conservation, the case may vary in non-coding elements, which may become a discrepancy in disease modeling [82]”.

An initial bias can be observed from the selection of the database as KEGG, as each contains different representations of the same biological pathway which can be seen represented as a variable result of the enrichment analysis and the contextualization thereof [83]. One should also consider the frequency of platform updates that may not yet cover newly discovered pathways or important updates related to emerging molecular mechanisms, thus truncating the ability for a comprehensive view of biological systems [84]”. (LINES 522 – 534).

Moreover, 3 references were included:

  1. Breschi, A.; Gingeras, T.R.; Guigó, R. Comparative transcriptomics in human and mouse. Nat. Rev. Genet. 2017, 18, 425-440.
  2. Mubeen, S.; Hoyt, C.T.; Gemünd, A.; Hofmann-Apitius, M.; Fröhlich, H.; Domingo-Fernández, D. The Impact of Pathway Da-tabase Choice on Statistical Enrichment Analysis and Predictive Modeling. Front. Genet. 2019, 10, 1203.
  3. Kanehisa, M.; Goto, S. KEGG: kyoto encyclopedia of genes and genomes. Nucleic Acids Res. 2000, 28, 27-30.

g) The authors have described the results section very well and have comprehensively presented bioinformatic analyses. Identification of specific pathways (e.g., focal adhesion, neurotrophin signaling) adds mechanistic insight into SCA7 pathology. But looking at the data the presentation of results seems overwhelming, with excessive details on GO terms and pathway components. This literally obscures the primary takeaways.

Answer: As the reviewer suggested, we have eliminated the overrepresentation of results for each miRNA analyzed by GO, which has led to the removal of some tables and figures. Therefore, in this version we show information that groups the 4 miRNAs for GO and KEGG. These findings are represented in the results section.

We hope its current form satisfices the reviewer.

h) The functional interpretation is lacking significantly, while enriched pathways are identified, their functional implications for SCA7 are not deeply discussed. The Figures and tables lack clarity and do not effectively summarize the findings.

Answer: Thanks to the reviewer for this observation. We understand your point of view; in order to clarify this issue, we have described the functional implications for SCA7 of the pathways in which there is evidence.

The following sentences were added in the discussion section:

It is important to consider that the pathways identified in our study and that will be described are related to pathways in other neurodegenerative diseases, so their partic-ipation and relevance is known. However, in SCA7 there are few findings of these pathways impacted by miRNAs hsa-miR-29a-3p, hsa-miR-132-3p, hsa-miR-25-3p and, hsa-miR-92a-3p. So they represent possible additional mechanisms for a better under-standing of the disease”. (LINES 343 – 348).

In silico pathways:

Adherens junction: LINES 355 - 377

Neurotrophin signaling: LINES 397 - 402

Endoplasmic reticulum processing: LINES 415 - 421

Actin cytoskeleton regulation: LINES 429 - 435

RNA transport: LINES 448 - 453

Apoptosis: LINES 454 - 468

Dopaminergic synapse: LINES 478 – 483

i) I felt that the figures are very much overlly representated. It can be shorted from 6 figures to 4 figures. The figures 3-6 looks same represenation of figures, if they have used bioinformatic approach then representing these data in different way would attract readers otherwise they will surely loose interest. It definitely needs to be changed.

Answer: We agree with the reviewer. We have revised the manuscript and the updated version now reflects these changes. We have removed the overrepresentation of results for each miRNA analyzed by GO, which led to the removal of four figures and four tables. In addition, we have modified the representation of the last four figures and reduced the number of figures from 6 to 4 in the main manuscript and now show 2 supplementary figures to improve clarity. We really appreciate your comments as they have helped us improve the presentation of our results.

Minor concerns

a) The discussion ties the results to broader neurodegenerative disease mechanisms. It also potentially links between miRNAs and pathological pathways like protein processing and actin cytoskeleton regulation are explored. But there are many assertions which seems to be speculative, with limited experimental evidence to support the proposed links between miRNAs and pathways. It often reiterates known findings without highlighting the unique contributions of this study. The authors might acknowledge the speculative nature of some claims and propose follow-up experiments to validate them. Also, they could emphasize novel insights from this study, such as the identification of specific hub genes or transcription factors.

Answer: Thanks to the reviewer for this valuable observation. We agree with your suggestion. We included a propose as perspective to follow-up experiments to functional validate in the discussion section (LINES 495 - 517).

As perspectives in the present study and considering the importance of achieving a better understanding of the pathways found with SCA7, we propose for the functional validation of the identified genes and their effects on the highlighted signaling pathways, ultimately contributing to the identification of molecules with potential therapeutic impact.

In this regard, studies using an in vitro cellular model of SCA7 could assess the overexpression or silencing of candidate miRNAs and monitor changes in the expression of their potential target genes using reverse transcription-polymerase chain reaction (RT-PCR). For example, the expression of Beta-catenin and E-cadherin in the focal ad-hesion pathway could be validated, as these molecules play a crucial role in cellular cohesion and intracellular communication. The loss of these functions could lead to apoptosis, ultimately resulting in the degeneration of retinal cells and neurons, as documented in both SCA7 patients and animal models.

Furthermore, if Beta-catenin and E-cadherin expression were regulated by any of the identified miRNAs, a decrease in these molecules would likely affect their function, thereby impacting the pathways in which they are involved. With this prior knowledge, it would be possible to explore whether these target molecules could have a therapeutic effect, considering their abundance, deficiency, and relevance in the affected tissue.

Additionally, an in vitro approach could validate the use of synthetic anti-miRNAs to block specific miRNAs, aiming to restore the function of their target genes and potentially influence disease progression. Conversely, if gene silencing were required, strategies such as adeno-associated virus (AAV)-mediated gene silencing via RNA interference or CRISPR-Cas could be employed”.

Also, we emphasize novel insights from this study such as the identification of specific hub genes or transcription factors.

In conclusion, unlike previous studies, we highlight through a bioinformatics approach core genes and transcription factors involved in different biological pathways and which are targets for the 4 miRNAs; which, in addition to being associated with neurodegenerative diseases, are also de-regulated in the plasma of patients with SCA7”. (LINES 27 - 31).

b) Tables lack summary statistics and clear rankings to prioritize findings.

Answer: Thanks again to the reviewer. In the revised manuscript, we added this information to prioritize findings.

c) Figures are way to dense and difficult to interpret. For example, the transcription factor-gene interaction networks (figure 3-6) are cluttered and lack explanatory captions.

Answer:  Following your recommendation, we have removed the overrepresentation of results for each miRNA analyzed by GO, which led to the removal of four figures and four tables. In addition, we have modified the representation of the last four figures and reduced the number of figures from 6 to 4 in the main manuscript and now show 2 supplementary figures (transcriptional factor-gene interaction) to improve clarity with attractive and explanatory way and we added explanatory titles.

Reviewer 3 Report (New Reviewer)

Comments and Suggestions for Authors

The stated goal of this study was to identify candidate genes that were targeted by 4 miRNAs that might play a role in spinocerebellar ataxia type 7 (SCA7), using a comprehensive in silico analysis comprising prediction of miRNA target genes, Gen Ontology enrichment and pathway analysis.  The selected miRNA genes were validated with transcriptome data from an SCA7 mouse model.  Presumably the for miRNAs chosen for analysis are included in a previous report of 71 differentially expressed miRNAs in patients with SCA7, but exactly why these four were chosen is not clarified in the introductory section of the communication.  Are these miRNA's of significantly higher differential expression in SCA7 patients, or was this choice based primarily on the known relationship of these miRNA's to neurodegenerative disorders in general?

The methods and results of this interrogation are appropriate and unexceptionable.  The analysis is thorough and utilizes state-of-the-art methods and produced a large quantity of target loci, as well as the usual array of figures in studies of this type.  However, the utility of this information with respect to advancing clinical information regarding SCA7 is questionable.

As parallel patient transcriptome data was not available the investigators chose to validate their results with expression arrays from cerebellar extracts of a mouse model (developmental days 10, 22 and 77) and in retinal cells from R7#-SCA7 transgenic mice at 3 and 9 weeks of development.  While the differentially expressed genes in these tissues overlapped with many of the predicted mRNA targets, this cannot be understood as anything other than coincidental.

This is a difficult communication to evaluate, as the methodology and results are excellent, but there is no specific outcome that suggests a clinical way forward.  Accordingly, I would rate the significance of this communication as only moderate. The lengthy discussion which highlights the relationship between many of the findings and their molecular relationships, but practically no relationship between the findings and clinical characteristics of SCA7 reifies this point.

Except for clarifying the selection of specific miRNAs to target, I have no worthwhile or short-term suggestions to improve the present communication, as I agree completely with the authors that additional biological analysis is required to advance the field.

Author Response

REVIEWER 3

We thank to the reviewer for the insightful comments and helpful criticisms. We have responded to all comments as detailed below and we now hope that you will find our revised Manuscript acceptable for publication in “Current Issues in Molecular Biology”.

REVIEWER'S COMMENTS:

He stated goal of this study was to identify candidate genes that were targeted by 4 miRNAs that might play a role in spinocerebellar ataxia type 7 (SCA7), using a comprehensive in silico analysis comprising prediction of miRNA target genes, Gen Ontology enrichment and pathway analysis.  The selected miRNA genes were validated with transcriptome data from an SCA7 mouse model. 

1. Presumably the for miRNAs chosen for analysis are included in a previous report of 71 differentially expressed miRNAs in patients with SCA7, but exactly why these four were chosen is not clarified in the introductory section of the communication?.

Answer: We included this valuable information according to the reviewer’s suggestion in a in the introduction section (LINES 76 - 104). In addition, we now clearly describe in “Material and Methods” section (2.1 Exploration of miRNA families and selection of miRNAs for the study) a detailed description of miRNAs selection criteria. (LINES 127 – 143).

We hope its current form satisfices the reviewer.

2. Are these miRNA's of significantly higher differential expression in SCA7 patients, or was this choice based primarily on the known relationship of these miRNA's to neurodegenerative disorders in general?

Answer: We approached this meaningful recommendation in the revised Manuscript. Yes, these miRNAs were choice in both cases, considering their higher differential expression in SCA7 patients in our previous study. In addition, these miRNAs have been significantly implicated in different neurodegenerative disorders including, neurogenesis, neuroinflammation and cognitive processes.

We now include brief information of the context in order to clarify the importance of selecting these miRNAs in the introduction section (LINES 76 - 104) and the Material and Methods section (LINES 127 – 143).

3. The methods and results of this interrogation are appropriate and unexceptionable. The analysis is thorough and utilizes state-of-the-art methods and produced a large quantity of target loci, as well as the usual array of figures in studies of this type.

However, the utility of this information with respect to advancing clinical information regarding SCA7 is questionable?.

Answer:

The reviewer's point of view is interesting, considering that an in silico analysis refers to scientific discoveries that are made through bioinformatics simulation rather than biological studies. In this sense, the data obtained through our in silico analysis of these 4 miRNAs could be used to predict and provide novel information and specific study of in SCA7.

Moreover, in silico analysis is a valuable way to test the true potential of new methods during development. Therefore, advances that increase the complexity of data simulations will allow researchers to better evaluate new analytical methods.

In the revised version, we included a propose as perspective to follow-up experiments to functional validate in near future to obtain clinical information regarding SCA7 (LINES 495 - 517).

As perspectives in the present study and considering the importance of achieving a better understanding of the pathways found with SCA7, we propose for the functional validation of the identified genes and their effects on the highlighted signaling pathways, ultimately contributing to the identification of molecules with potential therapeutic impact.

In this regard, studies using an in vitro cellular model of SCA7 could assess the overexpression or silencing of candidate miRNAs and monitor changes in the expression of their potential target genes using reverse transcription-polymerase chain reaction (RT-PCR). For example, the expression of Beta-catenin and E-cadherin in the focal ad-hesion pathway could be validated, as these molecules play a crucial role in cellular cohesion and intracellular communication. The loss of these functions could lead to apoptosis, ultimately resulting in the degeneration of retinal cells and neurons, as documented in both SCA7 patients and animal models.

Furthermore, if Beta-catenin and E-cadherin expression were regulated by any of the identified miRNAs, a decrease in these molecules would likely affect their function, thereby impacting the pathways in which they are involved. With this prior knowledge, it would be possible to explore whether these target molecules could have a therapeutic effect, considering their abundance, deficiency, and relevance in the affected tissue.

Additionally, an in vitro approach could validate the use of synthetic anti-miRNAs to block specific miRNAs, aiming to restore the function of their target genes and potentially influence disease progression. Conversely, if gene silencing were required, strategies such as adeno-associated virus (AAV)-mediated gene silencing via RNA interference or CRISPR-Cas could be employed”.

4. As parallel patient transcriptome data was not available the investigators chose to validate their results with expression arrays from cerebellar extracts of a mouse model (developmental days 10, 22 and 77) and in retinal cells from R7#-SCA7 transgenic mice at 3 and 9 weeks of development. While the differentially expressed genes in these tissues overlapped with many of the predicted mRNA targets, this cannot be understood as anything other than coincidental.

Answer:

We agree with the reviewer's comment, hence if there is a coincidence the mouse transcriptome with our results. This may be due to the type of study we conducted, that is, a prediction. However, we consider these to be valuable results, since this type of research could be establishes the bases to probe the veracity of the proposed pathways or mechanisms that can be explored in the near future with functional tests.

In this sense, we recognize this issue as limitations in our study in this moment. The following sentences were added in the discussion section (LINES 518 – 534).

We recognize that our study has some limitations. First, while enrichment analysis, based on statistical associations, highlighted potential molecular mechanisms involved in SCA7, further functional investigations are essential to confirm their relevance to the disease's pathobiology.

Second, it is essential to recognize that the interspecific differences between mouse models and humans may affect the study’s findings, this could be due to factors such as regulatory differences, like alternative splicing, chromatin structure, and enhancer activity, among others; that can alter expression patterns in mice and humans. Furthermore, while many of the protein-coding genes possess a high level of conservation, the case may vary in non-coding elements, which may become a discrepancy in disease modeling [82].

Third, an initial bias can be observed from the selection of the database as KEGG, as each contains different representations of the same biological pathway which can be seen represented as a variable result of the enrichment analysis and the contextualization thereof [83]. One should also consider the frequency of platform updates that may not yet cover newly discovered pathways or important updates related to emerging molecular mechanisms, thus truncating the ability for a comprehensive view of biological systems [84]”.

We hope its current form satisfices the reviewer.

5. This is a difficult communication to evaluate, as the methodology and results are excellent, but there is no specific outcome that suggests a clinical way forward. Accordingly, I would rate the significance of this communication as only moderate.

Answer: We understand the reviewer's perspective on this point. Thus, an in silico analysis involves the prediction of possible pathways involved in the pathology in question, with the aim of studying them at an experimental level in subsequent works. In addition, it is important to carry out this type of studies in rare diseases such as SCA7, since there is a lack of information on these miRNAs regarding the etiological causes and therefore, of adequate therapeutic strategies.

In this sense, there are currently studies similar to ours in the literature, where in silico research is carried out on different pathologies, including neurodegenerative diseases, to integrate scientific tools that allow elucidating an understanding. Below we included some references:

Satarker, S.; Maity, S.; Mudgal, J.; Nampoothiri, M. In silico screening of neurokinin receptor antagonists as a therapeutic strategy for neuroinflammation in Alzheimer's disease. Mol. Divers 2022, 26, 443-466.

Khani-Habibabadi, F.; Askari, S.; Zahiri, J.; Javan, M.; Behmanesh, M. Novel BDNF-regulatory microRNAs in neurodegenerative disorders pathogenesis: An in silico study. Comput. Biol. Chem. 2019, 83, 107153.

Romano, G.L.; Platania, C.B.; Forte, S.; Salomone, S.; Drago, F.; Bucolo, C. MicroRNA target prediction in glaucoma. Prog. Brain Res. 2015; 220:217-40.

In order to clarify this issue, in the revised version we have described the functional implications that involved a clinical way for SCA7 of the pathways in which there is evidence.

The following sentences were added in the discussion section:

It is important to consider that the pathways identified in our study and that will be described are related to pathways in other neurodegenerative diseases, so their partic-ipation and relevance is known. However, in SCA7 there are few findings of these pathways impacted by miRNAs hsa-miR-29a-3p, hsa-miR-132-3p, hsa-miR-25-3p and, hsa-miR-92a-3p. So they represent possible additional mechanisms for a better under-standing of the disease”. (LINES 343 – 348).

In silico pathways:

Adherens junction: LINES 355 - 377

Neurotrophin signaling: LINES 397 - 402

Endoplasmic reticulum processing: LINES 415 - 421

Actin cytoskeleton regulation: LINES 429 - 435

RNA transport: LINES 448 - 453

Apoptosis: LINES 454 - 468

Dopaminergic synapse: LINES 478 – 483

6. The lengthy discussion which highlights the relationship between many of the findings and their molecular relationships, but practically no relationship between the findings and clinical characteristics of SCA7 reifies this point.

Answer:

Considering the reviewer's suggestion, we expanded this point throughout the discussion for each pathway and as also shown in answer 5. (LINES 343 – 348).

In silico pathways:

Adherens junction: LINES 355 - 377

Neurotrophin signaling: LINES 397 - 402

Endoplasmic reticulum processing: LINES 415 - 421

Actin cytoskeleton regulation: LINES 429 - 435

RNA transport: LINES 448 - 453

Apoptosis: LINES 454 - 468

Dopaminergic synapse: LINES 478 – 483

7. Except for clarifying the selection of specific miRNAs to target, I have no worthwhile or short-term suggestions to improve the present communication, as I agree completely with the authors that additional biological analysis is required to advance the field.

Answer:

We thank to the reviewer for this suggestion. In the revised version, we included this valuable information in the introduction section (LINES 76 - 104). Moreover, it was described in the section: 2.1. Exploration of miRNA families and selection of miRNAs for the study (LINES 127 – 143).

This manuscript is a resubmission of an earlier submission. The following is a list of the peer review reports and author responses from that submission.

Round 1

Reviewer 1 Report

Comments and Suggestions for Authors

Dear Authors, the aim of your study is interesting and of clinical importance; however, the employed methodology lacks many critical aspects that should be included to verify your inference from this preliminary analysis. If abnormalities in the ATXN7 gene are important in SCA7, and if your choice was to investigate four miRNAs found deregulated in the plasma of SCA7 patients, then I think at least mutational landscape and transcriptomic data should have been included in the study to provide convincing results and conclusions. Functional analysis in the form of gene ontology and a few miRNA-target networks is not enough. A mention of patients and relevance of hsa-miR-29a-3p, hsa-miR-132-3p, hsa-miR-25-3p and hsa-miR-92a-3p is based on literature data. Please reconsider your methodology and include data on patients from databases such as TCGA, ICGC, and GEO. Investigate whether some changes in the expression profile based on RNA-Seq and miR-Seq are related to clinical data. Perform stratification if you find anything curious to gravitate towards ATXN7 and its alterations. Try to evaluate functional annotations using high-throughput examination or in vitro assays. Think about the biological meaning of the targets of the four miRNAs you selected from the literature - can they be somehow associated with ATXN7? Do they include known protein partners or upstream regulators (e.g., transcription factors) of ATXN7? Everything I mentioned would definitely enrich your study and a majority of them would most probably be required in such a study by experts in the field.

Author Response

REVIEWER 1

We thank all reviewers for their insightful comments and helpful criticisms. We have responded to all comments as detailed below and we now hope that you will find our revised Manuscript acceptable for publication in “Current Issues in Molecular Biology”.

Please reconsider your methodology and include data on patients from databases such as TCGA, ICGC, and GEO

Answer: We thank the reviewer for this opportune observation. We totally agree with respect to that the main limitation of this study is the lack of transcriptional analysis in humans, so this effort is a first approach. Therefore, we applied a statistical analysis, considering transcriptional data from an experimental model of SCA7 in vivo in transgenic mouse. For this purpose, we considered 2 data sets in GEO database GSE49115 and GSE3634, that were based on Agilent-014868 Whole Mouse Genome Microarray 4x44K G4122F and Affymetrix Mouse Expression 430A Array, respectively. The tissues analyzed in the models included the cerebellum and retina, which are the primary organs affected in SCA7. Cerebellar and retinal transcriptome genes (P value =0.05 and a Fold Change >1.5) were overlapped with miRNA target genes. After this correction, a new set of data was obtained that could be an approach to the role of miRNAs associated with SCA7. Corrected data are shown now throughout the manuscript and in Figure 6 (lines 321-331) and Figure 7 (lines 332-342). Additionally, the new analyses are described in detail in the Materials and Methods section. See Materials and Methods section “2.4. Validation of miRNA target genes in the SCA7 mouse” (lines 151-162).

Investigate whether some changes in the expression profile based on RNA-Seq and miR-Seq are related to clinical data.

Answer: As mentioned earlier, no human transcriptional data is available. However, following the insightful comments of the reviewer, we conducted an analysis based on deregulated genes identified in the SCA7 mouse model transcriptome using microarray studies. See Materials and Methods section “2.4. Validation of miRNA target genes in the SCA7 mouse” (lines 151-162) and “2.5. Transcription factors and hub genes” (lines 164-176). Through this approach, it was possible to identify key genes involved in the pathology via an independent KEGG pathway analysis to draw inferences. While further functional studies are needed to elucidate the biological relevance as shown in results “3.2. Identification of hub genes in KEGG pathways” (lines 266-289), “3.3. Validation of miRNA target genes in the mouse SCA7 transcriptome” (lines 291-308), and “3.4. Transcription factors and hub genes” (lines 343-374) with their respective figures: Figure 6 (lines 321-331), Figure 7 (lines 332-342), Figure 8 (lines 385-388) and Figure 9 (lines 389-392).

Perform stratification if you find anything curious to gravitate towards ATXN7 and its alterations.

Answer: We approached this meaningful recommendation in the revised Manuscript. Based on the ontological analysis, we identified pathways associated with the pathogenesis of SCA7 (see Figure 7C Apoptosis, line 335). Interestingly, PRKAR1A (a regulatory subunit of protein kinase A) has been reported to promote cell growth and division, as well as to regulate caspase 7 (CASP7), which is responsible for the proteolytic processing of ATXN-7 [56]. Notably, CASP7 is directly regulated by the miRNAs miR-132-3p and miR-29a-3p. Therefore, we have included a description of potential alterations involved in these processes in the "Discussion" section (lines 532- 537).

  1. Guyenet, S. J.; Mookerjee, S. S.; Lin, A.; Custer, S. K.; Chen, S. F.; Sopher, B. L.; La Spada, A. R.; Ellerby, L. M. Proteolytic cleavage of ataxin-7 promotes SCA7 retinal degeneration and neurological dysfunction.  Mol. Genet. 2015, 24, 3908-3917.

Try to evaluate functional annotations using high-throughput examination or in vitro assays.

Answer: We are conscious that additional studies are needed to support the importance of the SCA7-related miRNAs described in this study. As an attempt to reinforce our findings, we now analyzed in the revised Manuscript all potential human target genes of the relevant miRNAs that showed matches with those identified in the mouse transcriptome highlight, interestingly, at least 31 common target genes were identified. See Results in Figure 6 (lines 321-331) and Figure 7 (lines 332-342) and its relevance addressed in Discussion. This validation was followed by the identification of key genes within each KEGG pathway, as well as the analysis of potential transcription factors regulating these key genes. See Results in Figure 8 (lines 385-388) and Figure 9 (lines 389-392) and its relevance addressed in Discussion. Additional validation of identified miRNAs in a cell-based model of SCA7 is currently in progress and will be part of a separate Manuscript.  However, we briefly discuss in the discussion section the need to validate these results in experimental models for SCA7 as shown in Discussion section (lines 556-572). We hope these new results satisfy the reviewer.  

Think about the biological meaning of the targets of the four miRNAs you selected from the literature - can they be somehow associated with ATXN7?

Answer: Following this opportune observation, meaning of four miRNAs analyzed in this study is now included in the revised Manuscript, see in Introduction at lines 79-92. We include a brief discussion of hsa-miR-132-3p and hsa-miR-29a-3p and their potential role on the CASP7 regulation (see in “Discussion” lines 414 -418 and lines 532- 537).

Reviewer 2 Report

Comments and Suggestions for Authors

The manuscript entitled ‘In silico identification of pathways regulated by relevant miRNAs in spinocerebellar ataxia type 7’ written by Verónica Marusa Borgonio-Cuadra et al. presents interesting results regarding the role of four selected miRNAs in spinocerebellar ataxia type 7 (SCA7). The authors used previously obtained data to select miRNAs for in silico analysis that included exploration of target genes of these miRNAs and identification of associated molecular pathways in the context of the pathogenesis of SCA7. miRNA-gene regulatory networks were further created to illustrate the obtained associations. Obtained results have shown that altered regulation of miRNAs can potentially contribute to the mechanisms involved in the SCA7 progression and onset, and could serve as a diagnostic and therapeutic markers. However, the performed study is based on the limited methodology, and the study design is not sufficiently explained and justified. Therefore, some important improvements should be introduced to make the manuscript clear, scientifically sound, reproducible, and of greater importance to the scientific community. The English language is clear, but could be more academic.

General concept comments

   1.      In the Introduction section, the molecular mechanisms underlying SCA7, including signaling pathways, molecular processes, transcriptional regulatory mechanisms, genes and proteins involved, should be described in more detail to help the readers familiarise themselves with processes involved in this disease, and to provide the complete background of this topic. The extended background would provide an up-to-date overview of this topic and identify gaps in the knowledge.

   2.      The study design is not clear and must be justified in more detail. The authors selected four miRNAs (hsa-miR-29a-3p, hsa-miR-132-3p, hsa-miR-25a-3p, and hsa-miR-92a-3p) for analysis based on their previous work [15], but in that work another four miRNAs were identified as SCA7 biomarkers: hsa-let-7a-5p, hsa-let7e-5p, hsa-miR-18a-5p, and hsa-miR-30b-5p. Therefore, the criteria for using other miRNAs must be specified in detail in the Materials and Methods section, maybe together with a brief mention of results from this previous work (e.g., short description of compared groups, number of individuals in groups, fold change, and p values). Clarification of this discrepancy is crucial for the proper justification of the study design.

   3.      The presented work is grounded only on computational methods; therefore, validation of the obtained results (miRNA-gene regulation, cellular effects) in biological samples from SCA7 patients is absolutely necessary to confirm the clinical importance of the indicated markers. However, the presented results give a valuable preliminary view of the genetic markers of SCA7.

4.      The functional analysis performed by the authors includes only KEGG terms, but, as a standard, a Gene Ontology (GO) analysis should also be performed, and the most relevant functional terms should be provided, especially, that this type of analysis is also available in miRNet 2.0 tool. I suggest the authors to move figures 1-4 to the Supplementary Material and in their place provide results of Gene Ontology analysis.

5.      The authors provided miRNA-gene networks separately for the most relevant KEGG terms; however, the results could be significantly enriched by identifying hub genes in the network of all target genes and by presenting the subnetwork containing obtained hub genes and targeted miRNAs. The CytoHubba plugin for Cytoscpe, or another method, can be used for this purpose. Such results will help identify and visualise the most important target genes that have the highest potential to be confirmed in the validation studies.

6.      The main limitations of the study should be mentioned in the Discussion section.

I believe that my suggestions will help the authors improve the quality of their manuscript.

Author Response

REVIEWER 2

We thank all reviewers for their insightful comments and helpful criticisms. We have responded to all comments as detailed below and we now hope that you will find our revised Manuscript acceptable for publication in “Current Issues in Molecular Biology”.

The manuscript entitled ‘In silico identification of pathways regulated by relevant miRNAs in spinocerebellar ataxia type 7’ written by Verónica Marusa Borgonio-Cuadra et al. presents interesting results regarding the role of four selected miRNAs in spinocerebellar ataxia type 7 (SCA7). Therefore, some important improvements should be introduced to make the manuscript clear, scientifically sound, reproducible, and of greater importance to the scientific community.

  1. In the Introduction section, the molecular mechanisms underlying SCA7, including signaling pathways, molecular processes, transcriptional regulatory mechanisms, genes and proteins involved, should be described in more detail to help the readers familiarize themselves with processes involved in this disease, and to provide the complete background of this topic. The extended background would provide an up-to-date overview of this topic and identify gaps in the knowledge.

Answer:  Following this opportune recommendation, we now include a brief description of molecular mechanisms associated with SCA7 pathogenesis. Certainly, the additional information included in the current version will provide a better perspective on the mechanistic generalities of SCA7, enabling the reader to better understand the objectives of the present study included in the Introduction section.

  1. The study design is not clear and must be justified in more detail. The authors selected four miRNAs (hsa-miR-29a-3p, hsa-miR-132-3p, hsa-miR-25a-3p, and hsa-miR-92a-3p) for analysis based on their previous work [15], but in that work another four miRNAs were identified as SCA7 biomarkers: hsa-let-7a-5p, hsa-let7e-5p, hsa-miR-18a-5p, and hsa-miR-30b-5p. Therefore, the criteria for using other miRNAs must be specified in detail in the Materials and Methods section, maybe together with a brief mention of results from this previous work (e.g., short description of compared groups, number of individuals in groups, fold change, and p values). Clarification of this discrepancy is crucial for the proper justification of the study design.

Answer: We appreciate your comments and suggestions, we have briefly described the pertinent background related to the study of miRNA profiles evaluated in patients with SCA7. See fifth paragraph of Introduction section (lines 79-92). In addition, we addressed this point in Material and Methods with a brief description of the selection of miRNAs analyzed (See Materials and Methods section “2.1. Exploration of miRNA families and selection of miRNAs for the study” (lines 107-124). We hope this information clarifies the identification of the potential role of relevant miRNAs associated with SCA7

  1. The presented work is grounded only on computational methods; therefore, validation of the obtained results (miRNA-gene regulation, cellular effects) in biological samples from SCA7 patients is absolutely necessary to confirm the clinical importance of the indicated markers. However, the presented results give a valuable preliminary view of the genetic markers of SCA7.

Answer: We acknowledge the necessity of further studies to substantiate the significance of the SCA7-related miRNAs identified in this research. To strengthen our findings, we have included in the revised manuscript a comprehensive bioinformatic analysis of all potential human target genes of the relevant miRNAs, identifying at least 31 common target genes that overlap with those found in the mouse transcriptome, see Results in Figure 6 (lines 321-331), Figure 7 (lines 332-342) and their relevance addressed in Discussion. This analysis was complemented by the identification of key genes within each KEGG pathway and the examination of potential transcription factors regulating these key genes, see Figure 8 (lines 385-388) and Figure 9 (lines 389-392) and their relevance addressed in Discussion. Additionally, we have included a brief discussion on the necessity of validating these results in experimental models for SCA7 within the Discussion section (see lines 556- 572). It is worth mentioning that the validation of the identified miRNAs is currently underway using a cell-based model of SCA7, with these findings to be presented in a separate manuscript. However, we trust that these new results address the reviewer's concerns.

  1. The functional analysis performed by the authors includes only KEGG terms, but, as a standard, a Gene Ontology (GO) analysis should also be performed, and the most relevant functional terms should be provided, especially, that this type of analysis is also available in miRNet 2.0 tool. I suggest the authors to move figures 1-4 to the Supplementary Material and in their place provide results of Gene Ontology analysis.

Answer: Following this meaningful observation, we have included an analysis of Gene Ontology (see Material and Methods section “2.2. Gen Ontology and pathways enrichment analysis” at lines 126- 138). Corrected data are shown now throughout the manuscript and in new Figures 1 (lines 194-197), Figure 2 (lines 207-210), Figure 3 (lines 224-227) and Figure 4 (lines 240-243). In addition, we have moved Figures 1-4 from the first version to supplementary figures as suggested by the reviewer, as shown in Supplementary Materials (lines 600-606).

  1. The authors provided miRNA-gene networks separately for the most relevant KEGG terms; however, the results could be significantly enriched by identifying hub genes in the network of all target genes and by presenting the subnetwork containing obtained hub genes and targeted miRNAs. The CytoHubba plugin for Cytoscpe, or another method, can be used for this purpose. Such results will help identify and visualise the most important target genes that have the highest potential to be confirmed in the validation studies.

Answer: We totally agree with this observation. In the current version of the manuscript, we have presented the hub genes within the network of all target genes, as well as the subnetwork comprising the identified hub genes and their corresponding target miRNAs. This analysis was performed using the CytoHubba method in Cytoscape, as seen in Material and Methods section “2.3. Identification of hub genes in KEGG pathways” (lines 140-147), Figure 6 (lines 321-331) and Figure 7 (lines 332-342), as appropriately suggested by the reviewer.

  1. The main limitations of the study should be mentioned in the Discussion section.

Answer: Following this pertinent suggestion, we have included a conscientious discussion of the limitations of the study (see Discussion, penultimate paragraph at lines 556-572).

Round 2

Reviewer 2 Report

Comments and Suggestions for Authors

I appreciate the efforts of the authors to respond to my comments. I have only some minor remarks regarding the revised manuscript. I suggest the authors to perform Gene Ontology analysis simultaneously for all four miRNAs, similarly to KEGG analysis (Figure 5), and show one figure instead of figures 1-4. Figures 1-4 could be moved to Supplementary Material. Furthermore, gene symbols in figures 6-9 are not readable and should be enlarged. There are also some typos in the text (ex. lines 88, 115, 138, 202); therefore, I suggest the authors to check the text once again. I have no further comments.

Author Response

REVIEWER 2

We thank to the reviewer for their insightful comments and helpful criticisms. We have responded to all comments as detailed below and we now hope that you will find our revised Manuscript acceptable for publication in “Current Issues in Molecular Biology”.

Reviewer's comments

I appreciate the efforts of the authors to respond to my comments. I have only some minor remarks regarding the revised manuscript.

  1. I suggest the authors to perform Gene Ontology analysis simultaneously for all four miRNAs, similarly to KEGG analysis (Figure 5), and show one figure instead of figures 1-4. Figures 1-4 could be moved to Supplementary Material.

 Answer: Following this opportune recommendation, we now include Gene Ontology analysis simultaneously for all four miRNAs (Figure 1). Moreover, the initial 4 figures were moved to supplementary material as supplementary figures 1-4.

  1. Furthermore, gene symbols in figures 6-9 are not readable and should be enlarged. There are also some typos in the text (ex. lines 88, 115, 138, 202); therefore, I suggest the authors to check the text once again. I have no further comments.

 Answer: We thank the reviewer for his valuable comments and in this regard, I inform you, that they have been addressed.
